# General practitioners' everyday clinical decision-making on psychosocial problems of children and youth in the Netherlands

Lennard T. van Venrooij [1,2]*, Pieter C. Barnhoorn[2], Anne Marie Barnhoorn-Bos[3], Robert R. J. M. Vermeiren[1], Matty R. Crone[2]

1 Department of Child and Adolescent Psychiatry, LUMC Curium, Leiden, The Netherlands, 2 Department of Public Health and Primary Care, Leiden University Medical Center (LUMC), Leiden, The Netherlands, 3 GGZ Rivierduinen, Leiden, The Netherlands

* L.T.van_Venrooij@lumc.nl

## Abstract

### Background

Psychosocial problems in children and youth are common and may negatively impact their lives and the lives of their families. Since general practitioners (GPs) play a crucial role in detecting and intervening in such problems, it is clinically necessary to improve our insight into their clinical decision-making (CDM). The objective of this study was to explore which mechanisms underlie GPs' everyday CDM and their options for management or referral.

### Material and methods

This was a mixed methods study in which qualitative (interview substudy) and quantitative (online survey substudy) data were collected from GPs. Using a question framework and vignettes representative of clinical practice, GPs' CDM was explored. GPs were selected by means of an academic research network and purposive sampling. Data collection continued in constant comparison between both substudies. Using grounded theory, data from both substudies were triangulated into a flowchart consisting of mechanisms and management/referral options.

### Results

CDM-mechanisms were divided into three groups. GP-related mechanisms were GPs' primary approach of the problem (somatically or psychosocially) and their self-assessed competence to solve the problem based on interest in and knowledge about youth mental health care. Mechanisms related to the child and its social context included GPs' assessment whether there was psychiatric (co)morbidity, their sense of self-limitedness of the problem and assessed complexity of the problem. Whether GPs' had existing collaboration agreements with youth care providers and how they experienced their collaboration were collaboration-related mechanisms.

**Data Availability Statement:** All relevant data are within the paper and its Supporting Information files.

**Funding:** The author(s) received no specific funding for this work.

**Competing interests:** The authors have declared that no competing interests exist.

## Conclusion

The current study contributes to a relatively unexplored research area by revealing GP's in-depth thought processes regarding their CDM. However, existing research in this area supports the identified CDM mechanisms. Future initiatives should focus on validating CDM mechanisms in a larger population. If confirmed, mechanisms could be integrated into GP training and may offer guidelines for regulating proper access to mental health care services.

## Introduction

General practitioners (GPs) have an important role in identifying and managing appropriate help for children and youth with psychosocial problems [1]. In the literature, psychosocial problems are broadly described as impairments, activity limitations and participation restrictions related to mood and living, financial and domestic conditions and interpersonal relationships [2]. While psychosocial problems among children and youth are of frequent occurrence in general practice, GPs experience several barriers to their identification and management [1, 3]. Barriers relate to the consulting child and its family, to the GP and their methods and to referral to youth care providers [1, 4, 5]. Children and youth are often reluctant to disclose psychosocial problems, and problems in the abovementioned areas are often preceded by a long patient delay [5]. Furthermore, GPs often feel ill-equipped in their clinical decision-making (CDM) with respect to clinical training, communication skills and spendable consultation time, regardless of their age and work experience [1, 4, 6]. Last, a lack of referral options due to minimal community-based resources is also much reported by GPs as a barrier to their CDM [5, 6].

While the existing literature gives an overview of what can hamper GPs in their detection of and intervention in psychosocial problems in children and youth, it does not provide specific insight into their everyday CDM process [1, 4]. This process refers to mechanisms regarding 1) identification and diagnosis of a child or adolescent with psychosocial problems and/or 2) managing these problems, e.g., referral to outpatient mental health care services or additional psychosocial services [5, 7, 8]. Because of the gap in the literature, further exploring CDM in GPs is of primary importance. Only then we will be able to recognize mental health risk timely and accurately, and thus improve treatment and likely also a child's future.

Research shows that less than two-thirds of young people with mental health problems and their families access any professional help, suggesting a considerable level of unmet need among children and adolescents [9]. One study reports that just over a third (35.0%) of 4–17 year olds with a mental health disorder had seen a GP [10]. Psychosocial problems that are not identified or treated in time may lead to loss of mental, physical and social-educational well-being in children, possibly continuing into adulthood [11, 12]. Additionally, parents may develop feelings of failure, guilt and overburdening while providing care for their child [13].

To avert these consequences, the current study aims to explore the mechanisms for GPs' everyday CDM regarding psychosocial problems in children and youth. Furthermore, this study aims to assist GPs in their CDM and help them balance access to mental health care services (Fig 1 shows a simplified overview of the Dutch youth care system in relation to the social and medical domains concerning a child and its family and/or social network) [14, 15]. We explored the options for management or referral that GPs consider in common problem situations, as well as the facilitating and impeding mechanisms that influence their decision-making. To achieve these study goals, we used clinical vignettes. These have been used to measure provider attitudes toward various forms of medical care and are capable of reflecting the relationship between particular patient characteristics and providers' actual CDM [16].

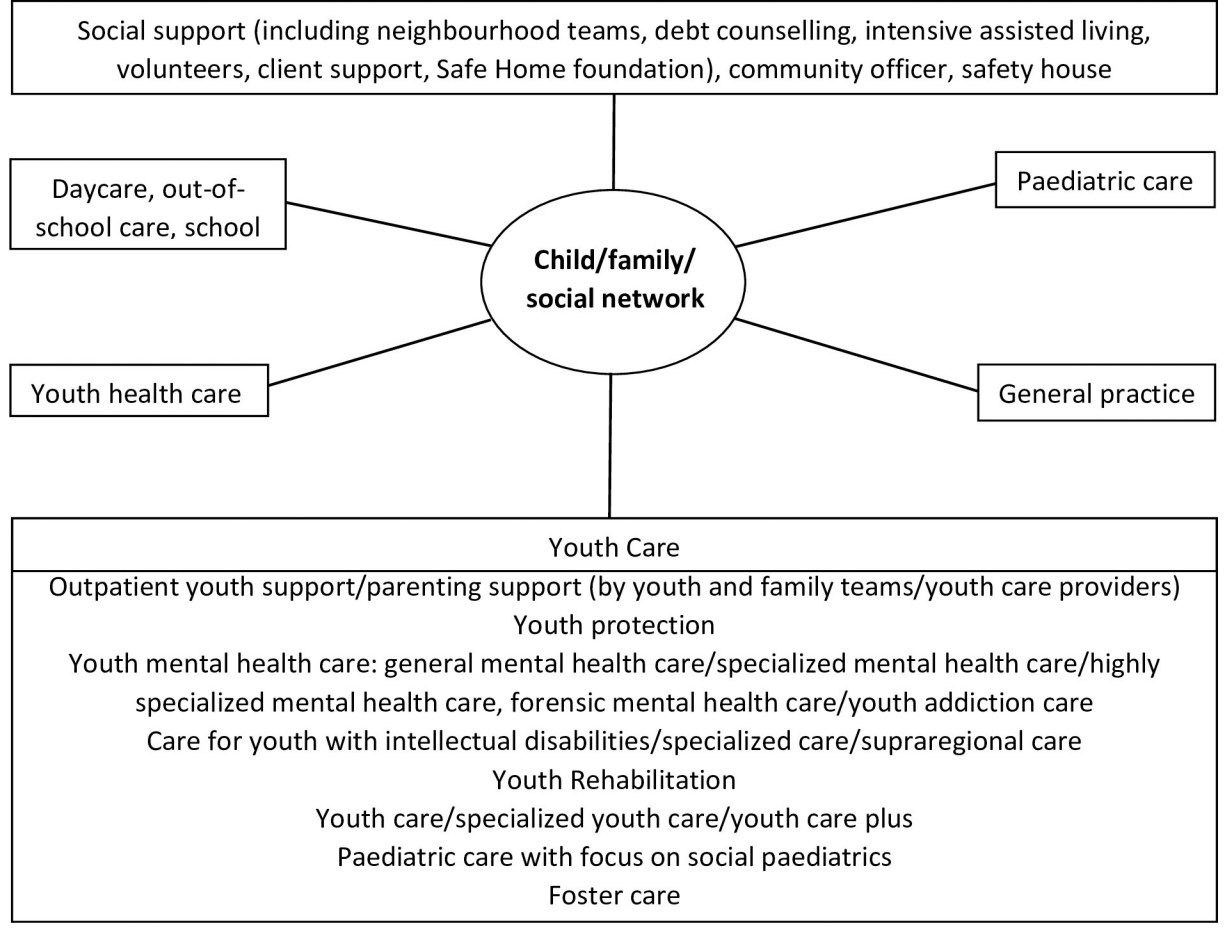

**Fig 1. Simplified overview of the Dutch youth care system in relation to the social and medical domains concerning a child and its family and/or social network[1].** [1]Figure derived from: Akwa GGZ [Internet]. Utrecht; c2022 [cited 2022 Feb 21]. Generieke module Samenwerkingsafspraken (jeugd); [about 9 screens]. Available from: https://www.ggzstandaarden.nl/generieke-modules/landelijke-samenwerkingsafspraken-jeugd-ggz/inleiding/doelstelling-van-deze-standaard (in Dutch).

## Materials and methods

### Study design

In this study, a complementary mixed methods design was used in which both qualitative (interview substudy) and quantitative (online survey substudy) data were collected from GPs working in the study region in order to identify CDM mechanisms. The results followed a convergent design and were analyzed independently and then integrated to develop a conceptual CDM flowchart using triangulation (Fig 2). The study was conducted in accordance with the COREQ checklist (COnsolidated criteria for REporting Qualitative research) [17].

**Interview substudy.** The interview substudy was conducted from February to June 2017 and comprised 30–45-minutes, one-to-one interviews among 14 GPs in their respective general practices. Interviews were planned 1–2 weeks in advance in consultation with a doctor's assistant. The interviews were semi-structured (i.e., open-ended questions followed by probes and transitions) [18, 19], were audio recorded using a voice recorder and were transcribed verbatim by LV. At the start of each interview, GPs were informed about the study's background and aim. Each GP was also asked to sign an informed consent form. For the interview substudy, 176 GPs were contacted by e-mail and telephone, of whom 14 were included.

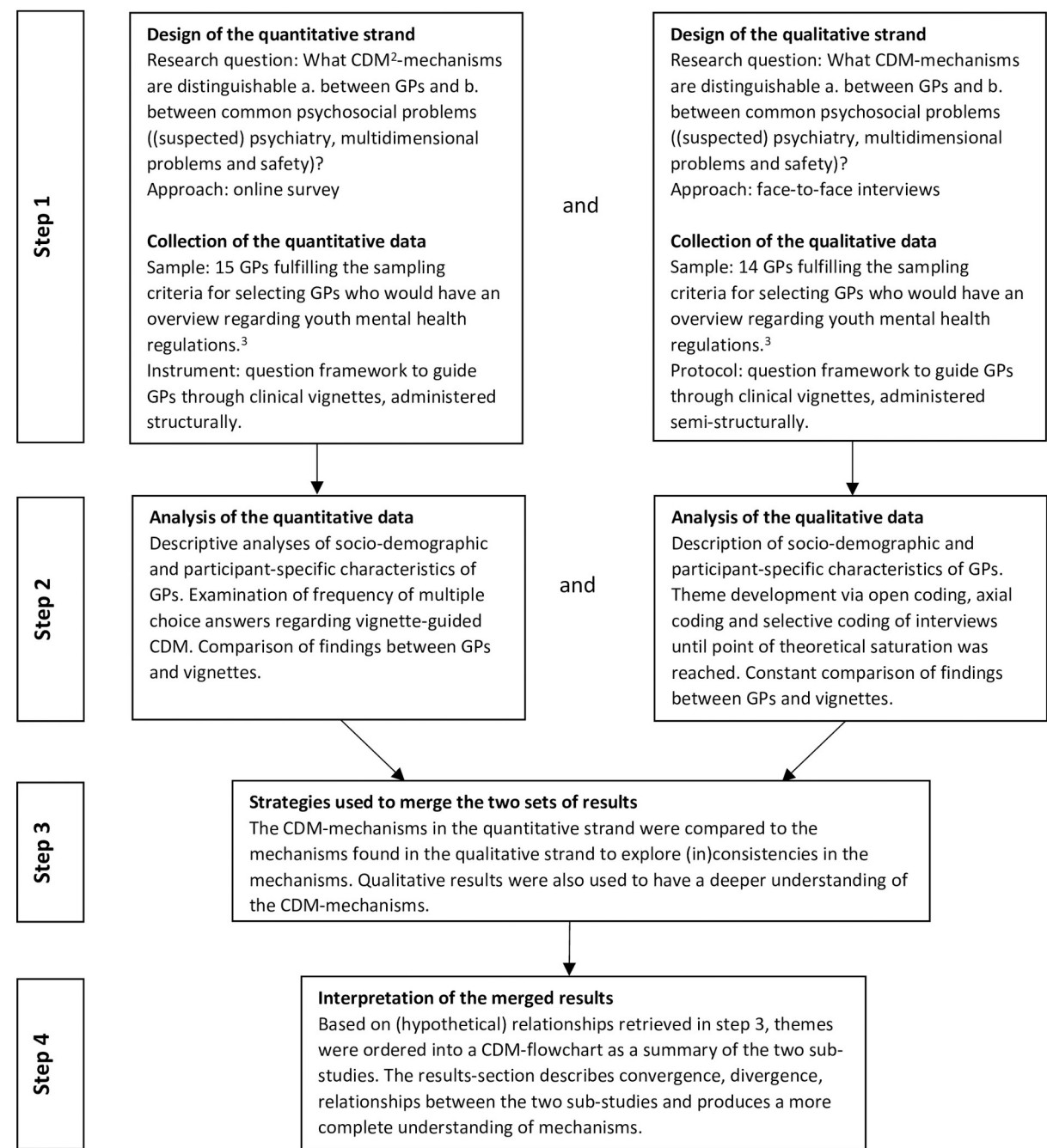

**Fig 2. Methods: Processes related to data acquisition and data analysis (convergent design)[1].** [1]Figure derived from: Creswell JW, Plano Clark VL. Choosing a mixed methods design. In: Creswell JW, Plano Clark VL, editors. Designing and conducting mixed methods research. Thousand Oaks: SAGE Publications Ltd; 2017. p. 53–106. [2]CDM = Clinical Decision-Making. [3]Sampling criteria included district in which GPs' general practices were established (with a maximum of one GP per district for the interview sub-study), GPs who reported seeing psychosocial problems among children and youth a minimum of three times per two weeks and experienced GPs who worked for themselves.

**Online survey substudy.** Quantitative data were derived from an online survey conducted in May 2017. The completion time of this questionnaire ranged from 10 to 20 minutes. The survey started with an introductory page including the study's background and aim, as well as a reference to the informed consent form that was attached to e-mails sent to each GP. The

online survey consisted of 27 questions based on a question framework. For the online survey substudy, 130 GPs were contacted, of whom 20 responded and 15 filled out the whole questionnaire.

## Research group

This study was carried out by a research group consisting of a GP (PB), a child psychologist (AB) who is also a member of a multidisciplinary family support team (youth and family team), a senior researcher in public health (MC) and a medical doctor-researcher (LV). Study-related tasks were divided between the research group members. Vignettes were formulated by PB and AB and approved by LV and MC. LV conducted all interviews alone and did most of the analyses. Weekly research meetings were scheduled with different combinations of the group for reflection on connections and patterns and for LV and other group members to discuss the study's progress. Ideas and hypotheses raised at the meetings were documented using memos.

## Ethics declarations

For this study, a medical ethical approval (P17.093) was granted by the medical ethical committee from Leiden University Medical Center (LUMC).

## Study setting

Both substudies were conducted in the Holland Rijnland region of the Netherlands which consists of 12 rural and 2 urban districts, each with its own regulations regarding youth mental health care provision. For recruitment, a regional academic research network (ELAN) was used in which 176 regionally established general practices were registered [20]. GPs registered in the research network had previously indicated an interest in scientific research. All GPs who were registered in the network were invited to participate in both the quantitative and qualitative substudy. All registered general practices were sent invitational e-mails with details of the substudies and contact details of the researchers. One reminder was sent for the interview study and three for the online survey.

**Participants.** GPs who indicated interest in participating in one or both substudies were selected based on purposive sampling [21]. Mandatory sampling criteria included the district in which GPs' general practices were established, with a maximum of one GP per district for the interview substudy. Since there are 14 districts in the Holland Rijnland region, the sample for the interviews initially has been restricted to n = 14. If two or more GPs from a particular district indicated their wish to participate in the qualitative substudy, the GP who responded first was included. Another sampling criterion was inclusion of experienced GPs who worked for themselves. Therefore, GPs were selected who would have an overview of youth mental health regulations, in order to gain a full picture of how youth mental health care provision is organized within the study region. In both the quantitative and qualitative substudy, the aim was to include GPs who see psychosocial problems among children and youths a minimum of three times per two weeks. However, during the recruitment, GPs often mentioned a lower frequency of once to thrice a month. We therefore decided to change this criterion and to also include GPs who reported to see these problems in a lower frequency. For the quantitative substudy, only GPs who filled out the whole questionnaire were included. Participation was voluntary. GPs who participated in the interviews received two small presents (a compass and a pen with a LUMC logo) in return for their participation. There was no compensation for participating in the online survey.

## Data gathering

**Clinical vignettes.** In both substudies, three fictional clinical vignettes—A "(suspected) psychiatry," B "multidimensional problems" and C "safety" (S1 Box)—were used to explore GPs' everyday CDM. Validation of vignettes was achieved through formulation of the vignettes by a child psychologist (AB) and a general practitioner (PB), based on personal clinical experience regarding referrals of children and youth to a local youth and family center. Also, the vignettes were verbally deemed recognizable with respect to clinical practice during several presentations to GPs and youth mental health workers outside the frame of this study [16, 22].

**Question framework.** For both substudies, the research group developed a question framework to explore CDM and to guide GPs through the clinical vignettes (see S2 Box). Unlike the interview study, the online survey included only multiple-choice questions. The question framework included questions on general demographics of the GP (i.e., name, gender, age, years working as a GP, years of working in current general practice), name of district he/she works in, type and frequency of encounters with psychosocial problems in children and youth, frequency of discussing psychosocial problems one on one with a mental health nurse practitioner (MHNP) and self-assessed knowledge of youth mental health care regulations. Questions regarding the clinical vignettes were about the first impression of the problem, thoughts on further diagnostics and management for solving the problem, plan for referral, recognizability of the vignettes with regard to practice, information relevant to achieving a plan for referral (e.g., patients', parents' and siblings' preferences, norms and values regarding treatment) and negative or positive collaboration experiences with other youth care providers (e.g., content, quality and speed of communication). The online survey concluded with three statements on financial cuts to youth mental health care services.

## Data analysis

Interview data were analyzed using grounded theory (Fig 2), comparing GPs' CDM between vignettes (within case) and between GPs (cross case). Using constant comparison, respondent validation and triangulation, the essential idea was to develop a single flowchart out of the two substudies in which all codes related to CDM of GPs were grouped into overarching family codes [23]. To find conceptual themes about CDM in the data, LV and PB separately applied codes to the first five interviews using open coding. LV continued this process until all interviews were open coded. During the research group sessions, researchers looked at the relationship between themes of interest by using axial coding and tried to find core themes using selective coding. LV continued these processes individually and reported intermediate findings to the research group members. Particular attention was paid to hypothetical causes and effects of CDM, e.g., GPs' self-perceived knowledge of youth mental health care services and their decision to refer or not refer. For the online survey, descriptive analyses were used to describe the sociodemographic and participant-specific characteristics of GPs and to examine the frequency of multiple-choice answers regarding vignette-guided CDM. Qualitative data were analyzed using ATLAS.ti© version 7.5 [24] and quantitative data using SPSS Statistics© version 24 [25].

## Results

### General findings

**Participants.** In total, 29 GPs (15 in the online survey, 14 in the interviews) were recruited, of whom 12 were female (5 in the online survey, 7 in the interviews). Unpurposely, all GPs who participated in the qualitative substudy did not fill out the questionnaire of the quantitative substudy. Therefore, there was no overlap between substudies with regards to

**Table 1. Characteristics of GPs–online survey (n = 15) and interviews (n = 14), N = 29.**

| Characteristics | Online survey | Interview study |
|---|---|---|
| Female sex–no. of GPs (%) | 5 (33%) | 7 (50%) |
| Number of years working as a GP–no. years | 18.6 years (range 5–38) | 18.7 years (range 3–33) |
| Number of years working in current general practice–no. of years | 15.7 years (range 2–36) | 13.0 years (range 3–32) |
| Working together with *mental health nurse practitioner* (MHNP)–no. of GPs (%) | 15 (100%) | 10 (71.4%) |
| • Yes, MHNP provides care for children and youths | 11 (73.3%) | 7 (50%) |
| • No, or GP works together with a MHNP but MHNP does not provide care for children and youths | 4 (26.7%) | 3 (21.4%) |
| Frequency of encountering psychosocial problems in children and youths during office hours–no. of GPs (%) | | |
| • Daily encounters (2–3 times a week) | 9 (60%) | 3 (21.4%) |
| • Weekly encounters ( time a week) | 2 (13.3%) | 4 (28.6%) |
| • Monthly encounters (–3 times a month) | 4 (26.6%) | 6 (42.9%) |
| • Less than monthly encounters (< time a month) | 0 (0.0%) | 1 (7.1%) |
| Recognizability of vignettes with respect to clinical practice–number of times mentioned by GPs | | |
| • Vignette A ((suspected) psychiatry) | N/A* | 6 times |
| • Vignette B (multidimensional problems) | N/A | 3 times |
| • Vignette C (safety) | N/A | 8 times |

*N/A = Not specifically asked

participants. GPs had worked an average of 18.6 years (range 5–38 years) and 18.7 years (range 3–33 years) in the field in the online survey and interview study, respectively. Moreover, they worked 15.7 years (range 2–36 years) and 13.0 years (3–32 years) in their current general practice in the online survey and interview study, respectively. Eighteen GPs worked with a mental health nurse practitioner (MHNP) who provides care for children and youth. Nearly all GPs encountered psychosocial problems at least monthly. According to interviewed GPs, vignettes A and C were most recognizable with respect to clinical practice (Table 1). The online survey did not question recognizability of clinical vignettes.

## Organization of mechanisms

Mechanisms for GPs' everyday CDM were organized using a flowchart (see Fig 3). For clarification purposes, mechanisms were subdivided into three domains related to 1) the GP, 2) the child and its social context and 3) GPs' collaboration with other care providers, which are described in detail below. Throughout the results section, identified CDM mechanisms are described following the order of the flowchart. However, the order of CDM mechanisms per GP deviated slightly from the flowchart order (Fig 3). Per CDM mechanism, the supporting results of the online survey are described first, following by the results of the interviews.

## Mechanisms related to GP

**Preferred approach.** To obtain an overview of the child's functioning in different life domains which may have been impeded by the problem situation (e.g., disruptive behavior at home, school, leisure), all but one surveyed GPs would ask for the child's or the adolescent's

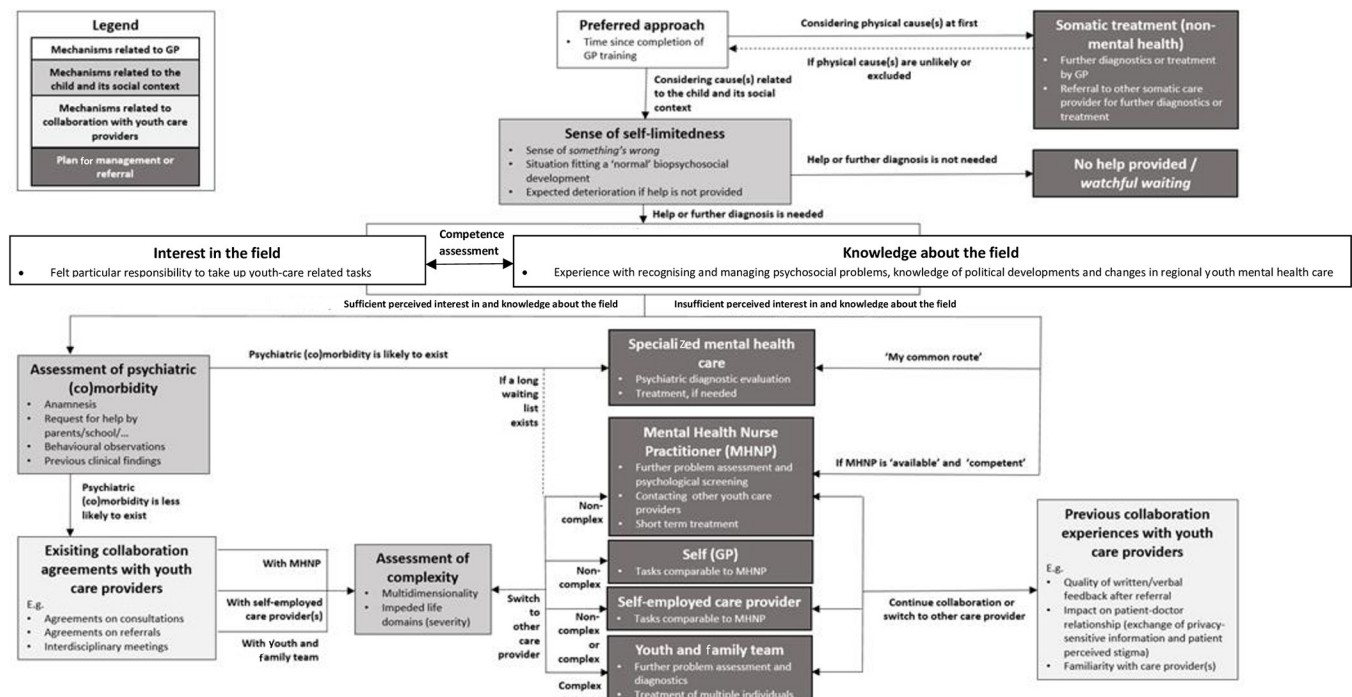

**Fig 3. Flowchart of mechanisms for GPs' everyday clinical decision-making when encountering psychosocial problems in children and youth[1,2].**
[1]Presented flowchart shows GPs' sequence of reflections and decisions when confronted with psychosocial problems in children and youths during office hours. [2]Boxes show in-depth considerations related to a specific mechanism.

opinion, as well as the parents' views of the problem situation. Less than half of the surveyed GPs would ask for another care provider's opinion and a youth health care provider's opinion.

Interviewed GPs mentioned contacting abovementioned persons to obtain more information but added that they would use this information to assess whether they attributed the presented problem(s) to somatic or psychosocial origins. A few interviewed GPs expressed their preference to arrange further physical diagnostic tests and, if needed, treatment by the GP themselves or by a somatic care provider, e.g., pediatric neurologist in vignette A. Presumed by the interviewer that physical cause(s) were unlikely or excluded, eventually all interviewed GPs considered psychosocial origins of the problem(s). After considering somatic and psychosocial origins of the problem, all interviewed GPs weighed whether the problem situation would solve itself without intervention or whether an intervention was necessary.

> *Quote 1B: "Mother has consulted me on her own, but I also think it would be interesting to hear what Sanne has to say."*

> *Quote 12A: "I would like to ask the child neurologist what causes his (Dave) symptoms, is it perhaps epilepsy?"*

**Competence assessment based on interest in the field.** The online survey contained no questions on perceived competence of the GP.

However, the interviews showed GPs' self-assessed competence to manage or refer the presented problem(s) was related to interest in the field on one hand and knowledge about the field on the other hand. A few interviewed GPs explicitly expressed feeling particularly responsible to take up youth-care-related tasks, having experience recognizing or treating

psychosocial problems. Other interviewed GPs did not mention anything about felt responsibility. A majority of GPs mentioned that identifying alarming problem situations and coordinating youth-care-related activities were perceived as their tasks, irrespective of their personal interest. Half of the interviewed GPs who explicitly expressed their interest in the field also participated in the district council to discuss recent political developments. A few GPs had previously participated in youth-care-related research. A small minority of GPs had done additional training in this area.

*Quote 2A: "My colleague's expertise is cutting and chopping, mine's communication."*

**Competence assessment based on knowledge about the field.**   The online survey showed some GPs have knowledge of recent developments in regional and national youth care, initiatives for interdisciplinary collaboration, regional referral options, legal regulations and money flows.

Interviewed GPs who claimed to have knowledge of changes in regional youth care provision stated they were kept well informed by their local authority about these changes. Following from the interviews, knowledge about the field also determined whether GPs first assessed (co)morbid mental health problems instead of choosing immediate referral to a child and youth psychiatrist. One interviewed GP called this immediate referral "my common route." In comparison, other GPs referred based on the specific request for help by the child, its parents or its school. If the GP was constrained by time or not well trained, a MHNP would be consulted to explore this request for help.

*Quote 9A*: *"I know that in this district youth care providers communicate with each other using an information loop."*

## Mechanisms related to the child and its social context

**Sense of self-limitedness.**   Throughout the vignettes, more than half of all surveyed GPs would advise parents to seek help, i.e., they thought further intervention was necessary.

The interviews revealed that this assessment of whether help or further diagnosis was needed was based on GPs' *gut feeling*. Interviewed GPs answered that this gut feeling was made up of a combination of factors, including: a sense of *something's wrong*, a problem situation not fitting GPs' expectations of a "normal" biopsychosocial development and expected deterioration if the GP refrained from any intervention, i.e., "watchful waiting."

*Quote 7C*: *"Well, I think this behavior is normal for her age."*

**Assessment of psychiatric (co)morbidity.**   The online survey contained no specific questions on the assessment of psychiatric (co)morbidity.

The interviews, however, showed that GPs assessed whether or not to consult a child and youth psychiatrist based on anamnesis, specific request for help, behavioral observations and previous clinical findings such as family history. However, due to long waiting lists, a few interviewed GPs chose to consult the youth and family team or a self-employed care provider instead, mainly guided by existing collaboration agreements with these youth care providers.

*Quote 4C*: *"It is somewhat unclear here, but Melany could be a troubled teenager with ADD or ADHD. If I would want further psychiatric diagnostic evaluation, I would refer to [name of a local mental health institution], instead of the youth and family center."*

**Assessment of complexity.**    The survey showed that in all three vignettes, GPs frequently thought about contacting a local youth and family team or asking their MHNP for further diagnosis and management. For vignette A, some surveyed GPs were also thinking about contacting a specialized mental health professional.

In the interviews, a majority of GPs reported that prior to selecting an intervention for further diagnosis or management, they assessed complexity of the vignettes. GPs spoke of a "complex" problem if the situation combined both multidimensionality and severity. A problem situation was called "multidimensional" if multiple individuals in a family were involved and multiple life domains were impeded. Furthermore, the degree to which a dimension was impeded was described as "severity." In noncomplex problem situations, most interviewed GPs would ask their mental health nurse practitioner (MHNP) or a self-employed child psychologist to provide short-term treatment or would consult other youth care providers. In more complex cases, GPs would contact a local youth and family team because these teams were considered to offer rapid social support to multiple individuals in a family at the same time. Furthermore, specialized mental health care services would have been chosen in more complex cases. If a consulted care provider would assess the problem(s) to be more or less complex than assessed by the GP, all interviewed GPs were prepared to refer to another youth care provider.

> *Quote 2B: "I don't think I have to refer every teenager who lives in a stressful home situation."*

> *Quote 3B: "The self-employed child psychologist I usually refer to also pays attention to the child's social system."*

## Mechanisms related to collaboration with youth care providers

**Existing collaboration agreements.**    The online survey contained no specific questions on the importance of existing collaboration agreements in GPs' CDM.

However, the interviews showed that, in cases where GPs thought psychiatric (co)morbidity seemed less likely, they inventoried which youth care provider they already had collaboration agreements with on, for example, consultations, referrals and interdisciplinary meetings. The most frequently mentioned youth care providers to be contacted were MHNPs, self-employed care providers and local youth and family teams.

> *Quote 5A: "Every six weeks, I speak to our MHNP about patients she has seen."*

> *Quote 14A: "The youth and family team of our district advises us on waiting lists."*

**Previous collaboration experiences.**    Surveyed GPs were asked to react to several statements regarding their previous collaboration experiences with youth care providers. Their answers showed a few GPs opining that youth and family teams are suitable for managing social-system-related problems and most GPs opining that budget cuts in youth care services have led to deteriorated quality of care.

The interviews provided more detail with regards to these answers. Almost all interviewed GPs thought recent budget cuts in youth care provision increased the possibility of having negative collaboration experiences rather than positive ones. Negative experiences included poor quality of written or verbal feedback after referral, previous referrals having a negative impact on patient-doctor relationships due to unsafe exchange of private information, patient-perceived stigma after referral, personal unfamiliarity with care provider(s) and low perceived

expertise. Negative collaboration experiences with a care provider—local youth and family teams in particular—resulted in referring to another care provider, even if the former provider might have provided more suitable help. Positive experiences made it more likely for GPs to refer to a self-employed care provider they preferred.

> **Quote 13A: "*After a lot of conversations (with mother of the child, among others), the person from the youth and family team concluded it was a difficult situation. That's a very thorough conclusion after six months.* \*laughs cynically\***"**

## Discussion

An online survey and an interview study yielded a flowchart containing mechanisms for GPs' everyday CDM regarding psychosocial problems in children and youth, resulting in a variety of options for management and referral. Identified mechanisms were subdivided into three domains related to 1) the GP, 2) the child and its social context and 3) the GPs' collaboration with other youth care providers. GP-related mechanisms included the preferred approach and perceived competence based on interest in and knowledge about the field. Mechanisms related to the child and its social context included assessed psychiatric (co)morbidity, sense of self-limitedness and assessed complexity of the presented problem(s). Existing collaboration agreements and previous collaboration experiences formed the last domain. With regards to GPs' management and referral when confronted with presented problem(s), consultation options varied between specialized mental health care services, a MHNP, a self-employed care provider, the local youth and family team and follow-up appointments with the GP themselves.

In comparison to the literature, this study contributes to a relatively unexplored research area, by providing data about the in-depth thought processes of GPs regarding their CDM. In line with previous studies, this study shows that psychiatric morbidity is commonly seen in general practice, often co-presenting with problems in other life dimensions [1]. Also, we found that some GPs give more priority than others to somatic instead of psychosocial problems, which may lead to referrals of children with psychiatric (co)morbidity to somatic care providers. This was especially the case with experienced GPs. This finding can be explained by literature reporting current medical training tends to focus on the patient in their social context compared to isolated medical problems [1, 4, 26]. Corresponding with previous studies, which report that experience, training and attitudes of GPs were key to the correct diagnosis of psychiatric disorders [26], the current study showed that GPs' sufficient perceived interest and knowledge about the field resulted in consideration of more CDM-mechanisms prior to referral instead of an immediate referral to specialized mental health care ('my common route'). However, our findings only refer to GPs' self-perceived competence and do not comprise measurement of actual skills. As for the recently installed youth and family teams at the time of this study, GPs refrained from considering and consulting these teams if they had negative collaboration experiences, even if this choice resulted in poorer quality of care. This finding corresponds with another study in which poor communication, trust and support resulted in perceived patient delay [27]. Our findings underline the importance of interprofessional collaboration as a key factor in initiatives designed to increase the effectiveness of health services offered to the public [28].

This study has several strengths. First, the mixed methods design and usage of vignettes made it possible to examine CDM mechanisms from multiple perspectives and in more detail compared to self-contained interview studies and online surveys [16]. Second, the vignettes were validated in a multidisciplinary research group and were deemed recognizable with

respect to clinical practice by GPs, also outside the frame of this study [22]. There are also limitations. First, present study provided some indication for different types of GPs regarding their CDM, as has also been described by Roberts et al 2014, who described the three role archetypes GPs can fulfill while identifying mental health problems in children and youths: 'fixers', 'future planners' and 'collaborators' [1]. However, the sample was not sufficiently large to be able to distinguish divergent types of CDM. Second, there is a possibility of self-selection bias. While characteristics of participating GPs were largely balanced due to use of purposive sampling, no information was retrievable regarding GPs who decided not to participate or who did not respond to the study invitation [29]. Furthermore, GPs who were particularly interested in youth health care might have been included, which may have influenced our results. There was no information retrievable regarding the 5 GPs who stopped filling out the questionnaire after the sociodemographic questions. However, the ones that stopped were not different from the GPs who filled out the whole questionnaire, so the authors think exclusion of these 5 GPs has little to no consequences with regards to our study results regarding GPs' CDM. Last, some mechanisms were explored to a higher degree in the interview substudy compared to the online survey substudy. Due to a restricted study time schedule, the survey questions were developed during data analysis of the interviews.

The authors suggest that future research is aimed at confirming or disproving the CDM mechanisms found, preferably in settings with multiple general practices. Also, it would be interesting to differentiate between different profiles of GPs based on their CDM in future studies. Since it is important that GPs address psychological problems in children and youth early, investments to improve their clinical practice regarding youth care are necessary, e.g., in medical education. Furthermore, more effective cross-disciplinary work should be encouraged, so that the expertise of multiple care providers can be utilized during GPs' assessment and decision whether or not to refer [3, 30]. GPs' daily CDM may be interesting to policy makers, so that usage of community-based resources by care providers and families can be well thought out [5]. The abovementioned initiatives should result in providing families with the help they need most.

## Conclusions

Participating GPs in a small, mixed methods vignette study showed three domains of CDM mechanisms for the GP, the child and its social context and the GPs' collaboration with other youth care providers. Future initiatives should focus on validating CDM mechanisms in a larger population. If confirmed by quantitative studies, mechanisms could be integrated into GP training and may offer guidelines for regulating proper access to mental health care services.

## Supporting information

**S1 Box. Vignettes—A '(suspected) psychiatry', B 'multidimensional problems' and C 'safety'.**
(DOCX)

**S2 Box. Interview guide.**
(DOCX)

**S1 File. Survey data [in Dutch].**
(XLSX)

**S2 File. Interviews pseudonymized data [in Dutch].**
(DOCX)

**S3 File. Interviews codes overview [in Dutch].**
(DOCX)

**S1 Fig. Interviews connections (family)codes and quotes [in Dutch].**
(TIF)

# Acknowledgments

We thank all participating GPs, the research group and everyone who made this study possible. Special thanks go to Ms. E. Wieling, Ms. E. Visser and Ms. A. Stelling for their organizational support.

# Author Contributions

**Conceptualization:** Lennard T. van Venrooij, Pieter C. Barnhoorn, Anne Marie Barnhoorn-Bos, Matty R. Crone.

**Data curation:** Lennard T. van Venrooij.

**Formal analysis:** Lennard T. van Venrooij, Pieter C. Barnhoorn, Matty R. Crone.

**Funding acquisition:** Pieter C. Barnhoorn.

**Investigation:** Lennard T. van Venrooij, Matty R. Crone.

**Methodology:** Lennard T. van Venrooij, Anne Marie Barnhoorn-Bos, Robert R. J. M. Vermeiren, Matty R. Crone.

**Resources:** Lennard T. van Venrooij, Pieter C. Barnhoorn, Anne Marie Barnhoorn-Bos.

**Supervision:** Pieter C. Barnhoorn, Anne Marie Barnhoorn-Bos, Robert R. J. M. Vermeiren, Matty R. Crone.

**Validation:** Pieter C. Barnhoorn, Anne Marie Barnhoorn-Bos, Matty R. Crone.

**Writing – original draft:** Lennard T. van Venrooij, Matty R. Crone.

**Writing – review & editing:** Pieter C. Barnhoorn, Anne Marie Barnhoorn-Bos, Robert R. J. M. Vermeiren, Matty R. Crone.

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
