## [Decision Letter · Decision Letter 0]

27 Aug 2022

PONE-D-22-10593General practitioners’ everyday clinical decision-making on psychosocial problems of children and youth in the NetherlandsPLOS ONE

Dear Dr. van Venrooij,

Thank you for submitting your manuscript to PLOS ONE. After careful consideration, we feel that it has merit but does not fully meet PLOS ONE’s publication criteria as it currently stands. Therefore, we invite you to submit a revised version of the manuscript that addresses the points raised during the review process.

Please note that we have only been able to secure a single reviewer to assess your manuscript. We are issuing a decision on your manuscript at this point to prevent further delays in the evaluation of your manuscript. Please be aware that the editor who handles your revised manuscript might find it necessary to invite additional reviewers to assess this work once the revised manuscript is submitted. However, we will aim to proceed on the basis of this single review if possible. The reviewer has raised some significant concerns regarding the reporting of this study. When revising your manuscript, please ensure you address each of the comments fully; the requested methodological detail has a bearing on whether the conclusions of this study can be reasonably found to add to the body of knowledge in this field.

We look forward to receiving your revised manuscript.

Kind regards,

Hugh Cowley

Staff Editor

PLOS ONE

Journal Requirements:

Reviewers' comments:

Reviewer's Responses to Questions

**Comments to the Author**

1. Is the manuscript technically sound, and do the data support the conclusions?

Reviewer #1: Partly

2. Has the statistical analysis been performed appropriately and rigorously? 

Reviewer #1: N/A

3. Have the authors made all data underlying the findings in their manuscript fully available?

Reviewer #1: No

4. Is the manuscript presented in an intelligible fashion and written in standard English?

Reviewer #1: Yes

5. Review Comments to the Author

Reviewer #1: Only very little is reported from the qualitative study, since most findings are presented as percentages, which is inappropriate given the non-representative character of both samples (see remarks below). I would assume that IF e.g. divergent types of CDM could be found, this would be of more relevance to the average reader, especially since the qualitative study did not seem to have informed the quantitative arm of the study substantially (both studies were conducted in the same timeframe).

Sampling technique: All GP´s were recruited from a regional academic research Network – this might have lead to selection effects (or all all GPs in the region members of this network)? I did not fully understand the criteria for GPs who were approached for the quantitatve study: all remaining GPs not expressing their interest in research? Plus GPs not responding to the qualittive study?

It seems that GPs could indicate their interest in both studies, but there was no overlap between both study groups – was this by purpose or not? Furthermore, if the qualitative study part included one participant max. per district, this would mean that the sample initially has been restricted to n=14 – or did the number and distribution happened by chance? If further inclusion/exclusion criteria like e.g. restricting the number of perticipants for the qualitative study based on the GP´s district, this should be made clear and should be justified.

The sample obtained showed a long history of working as GPs; it might be of interest to see the range of years they have been working in the field. One would expect that GP´s with an average history of 18 years of practive usually have more routine in their CDM, and this might in part explain their hight self-rated competence.

Inclusion criteria (“GPs reporting seeing psychosocial problems among children and youth a minimum of three times per two weeks”) and finding in table 1 (max monthly psychosocial problems in children (35%) in the qualitative study and 50% in the quantitative study do not match. Participants . Are the inclusion criteria defined as ALL criteria need to be fulfilled or at least one of the criteria has to be fulfilled?

6. PLOS authors have the option to publish the peer review history of their article (what does this mean?). If published, this will include your full peer review and any attached files.

Reviewer #1: No

---

## [Author Response · Author response to Decision Letter 0]

18 Oct 2022

Point-by-point response to comments given by the reviewer

* Only very little is reported from the qualitative study, since most findings are presented as percentages, which is inappropriate given the non-representative character of both samples (see remarks below).

Pages 13-19. The authors have now used words to describe the findings, instead of percentages (e.g. ‘some’, ‘many’, ‘a few’). Additionally, we have made a distinction between the qualitative and quantitative results by separating the two arms more clearly in the text (using indents). It was our intention to compare the quantitative with the qualitative data and to explore whether both substudies would show similar results.

* I would assume that IF e.g. divergent types of CDM could be found, this would be of more relevance to the average reader, especially since the qualitative study did not seem to have informed the quantitative arm of the study substantially (both studies were conducted in the same timeframe).

Page 21. We used a convergent mixed-method design, as our first aim was to explore what kind of decision-making processes could be distinguished. By combining and comparing the perspectives from qualitative and quantitative data, we aimed to triangulate the findings. Additionally, the qualitative findings explained to some extent the mechanisms behind the decision-making processes that were found in both study arms. 

The study provided some indication for different types of GPs regarding their CDM, but the sample was not sufficiently large to be able to distinguish divergent types of CDM. However, it will be an interesting next step to assess whether different profiles of GPs can be distinguished, as has also been described by Roberts et al 2014, who described the three role archetypes GPs can fulfill while identifying mental health problems in children and youths: ‘fixers’, ‘future planners’ and ‘collaborators’. We elaborated on this point in the Discussion.

* Sampling technique: All GP´s were recruited from a regional academic research Network – this might have lead to selection effects (or all all GPs in the region members of this network)? 

Page 21. We agree the recruitment from the regional academic research network might have led to inclusion of GPs who were particularly interested in youth care. This may have influenced our results. Therefore, we have added this point to the Discussion.

* I did not fully understand the criteria for GPs who were approached for the quantitatve study: all remaining GPs not expressing their interest in research? Plus GPs not responding to the qualittive study?

It seems that GPs could indicate their interest in both studies, but there was no overlap between both study groups – was this by purpose or not?

Pages 8 & 11. All GPs who were registered in the research network were invited to participate in both the quantitative and qualitative substudy. Unpurposely, all GPs who participated in the qualitative substudy did not fill-out the questionnaire of the quantitative substudy. Both substudies were conducted at roughly the same time. We have processed our response in the text.

* Furthermore, if the qualitative study part included one participant max. per district, this would mean that the sample initially has been restricted to n=14 – or did the number and distribution happened by chance? If further inclusion/exclusion criteria like e.g. restricting the number of participants for the qualitative study based on the GP´s district, this should be made clear and should be justified.

Page 8. Since there are 14 districts in Holland Rijnland, the sample initially has been restricted to n=14. If two or more GPs from a particular district indicated their wish to participate in the qualitative substudy, the GP who responded first was included. We clarified our response in the text describing our sampling criteria.

* The sample obtained showed a long history of working as GPs; it might be of interest to see the range of years they have been working in the field. One would expect that GP´s with an average history of 18 years of practive usually have more routine in their CDM, and this might in part explain their hight self-rated competence.

Pages 11, 20 and Table 1. We have reported the range of years GPs have been working in the field and the range of years they have been working in their current general practice. We did not found that GPs who worked more years in the field had a higher self-rated competence. However, we found that more experienced GPs tend to give more priority to somatic instead of psychosocial problems. 

* Inclusion criteria (“GPs reporting seeing psychosocial problems among children and youth a minimum of three times per two weeks”) and finding in table 1 (max monthly psychosocial problems in children (35%) in the qualitative study and 50% in the quantitative study do not match. 

Page 9. We agree this inclusion criteria and these findings cause some confusion. In both substudies, we aimed to include GPs who see psychosocial problems among children and youths a minimum of three times per two weeks. However, during the recruitment phase, there were quite some GPs who reported seeing psychosocial problems about once to three times a month. Therefore, we have adjusted the sampling criteria.

*Participants . Are the inclusion criteria defined as ALL criteria need to be fulfilled or at least one of the criteria has to be fulfilled?

Pages 8 & 9. The authors confirm that all sampling criteria needed to be fulfilled. As mentioned above, the criterion of seeing psychosocial problems among children and youths a minimum of three times per two weeks was found to be too strict, although it was our aim to include GPs who see these problems in a sufficient frequency.

---

## [Editor Report · Decision Letter 1]

6 Nov 2022

PONE-D-22-10593R1General practitioners’ everyday clinical decision-making on psychosocial problems of children and youth in the NetherlandsPLOS ONE

Dear Dr. Lennard van Venrooij

Thank you for submitting your manuscript to PLOS ONE. After careful consideration, we feel that it has merit but does not fully meet PLOS ONE’s publication criteria as it currently stands. Therefore, we invite you to submit a revised version of the manuscript that addresses the points raised during the review process.

REVIEWERS COMMENTS: Only very little is reported from the qualitative study, since most findings are presented as percentages, which is inappropriate given the non-representative character of both samples (see remarks below). I would assume that IF e.g. divergent types of CDM could be found, this would be of more relevance to the average reader, especially since the qualitative study did not seem to have informed the quantitative arm of the study substantially (both studies were conducted in the same timeframe).

Sampling technique: All GP´s were recruited from a regional academic research Network – this might have lead to selection effects (or all all GPs in the region members of this network)? I did not fully understand the criteria for GPs who were approached for the quantitatve study: all remaining GPs not expressing their interest in research? Plus GPs not responding to the qualittive study?

It seems that GPs could indicate their interest in both studies, but there was no overlap between both study groups – was this by purpose or not? Furthermore, if the qualitative study part included one participant max. per district, this would mean that the sample initially has been restricted to n=14 – or did the number and distribution happened by chance? If further inclusion/exclusion criteria like e.g. restricting the number of perticipants for the qualitative study based on the GP´s district, this should be made clear and should be justified.

The sample obtained showed a long history of working as GPs; it might be of interest to see the range of years they have been working in the field. One would expect that GP´s with an average history of 18 years of practive usually have more routine in their CDM, and this might in part explain their hight self-rated competence.

Inclusion criteria (“GPs reporting seeing psychosocial problems among children and youth a minimum of three times per two weeks”) and finding in table 1 (max monthly psychosocial problems in children (35%) in the qualitative study and 50% in the quantitative study do not match. Participants . Are the inclusion criteria defined as ALL criteria need to be fulfilled or at least one of the criteria has to be fulfilled?

We look forward to receiving your revised manuscript.

Kind regards,

Margaret Williams, Ph.D

Academic Editor

PLOS ONE

Additional Editor Comments (if provided):

Only very little is reported from the qualitative study, since most findings are presented as percentages, which is inappropriate given the non-representative character of both samples (see remarks below). I would assume that IF e.g. divergent types of CDM could be found, this would be of more relevance to the average reader, especially since the qualitative study did not seem to have informed the quantitative arm of the study substantially (both studies were conducted in the same timeframe).

Sampling technique: All GP´s were recruited from a regional academic research Network – this might have lead to selection effects (or all all GPs in the region members of this network)? I did not fully understand the criteria for GPs who were approached for the quantitative study: all remaining GPs not expressing their interest in research? Plus GPs not responding to the qualitative study?

It seems that GPs could indicate their interest in both studies, but there was no overlap between both study groups – was this by purpose or not? Furthermore, if the qualitative study part included one participant max. per district, this would mean that the sample initially has been restricted to n=14 – or did the number and distribution happened by chance? If further inclusion/exclusion criteria like e.g. restricting the number of participants for the qualitative study based on the GP´s district, this should be made clear and should be justified.

The sample obtained showed a long history of working as GPs; it might be of interest to see the range of years they have been working in the field. One would expect that GP´s with an average history of 18 years of practice usually have more routine in their CDM, and this might in part explain their hight self-rated competence.

Inclusion criteria (“GPs reporting seeing psychosocial problems among children and youth a minimum of three times per two weeks”) and finding in table 1 (max monthly psychosocial problems in children (35%) in the qualitative study and 50% in the quantitative study do not match. Participants . Are the inclusion criteria defined as ALL criteria need to be fulfilled or at least one of the criteria has to be fulfilled?
---

## [Author Response · Author response to Decision Letter 1]

13 Nov 2022

* Only very little is reported from the qualitative study, since most findings are presented as percentages, which is inappropriate given the non-representative character of both samples (see remarks below).

Pages 7, 9, 11-19 & 21. We have now used words to describe the findings, instead of percentages (e.g. ‘some’, ‘many’, ‘a few’). Additionally, we have made a distinction between the qualitative and quantitative results by separating the two arms more clearly in the text (using indents). It was our intention to compare the quantitative with the qualitative data and to explore whether both substudies would show similar results. 

In total, 14 GPs were interviewed. Next, 15 GPs completed the questionnaire. Five GPs started, but did not filled it out completely. This latter group did not differ greatly from the group that filled out the whole questionnaire regarding sociodemographic characteristics or frequency of seeing children and youths with psychosocial problems. In total, we had either qualitative or quantitative clinical decision-making information of 29 GPs, using the same vignettes in both substudies.

* I would assume that IF e.g. divergent types of CDM could be found, this would be of more relevance to the average reader, especially since the qualitative study did not seem to have informed the quantitative arm of the study substantially (both studies were conducted in the same timeframe).

Page 21. We used a convergent mixed-method design, as our first aim was to explore what kind of decision-making processes could be distinguished. By combining and comparing the perspectives from qualitative and quantitative data, we aimed to triangulate the findings. Additionally, the qualitative findings explained to some extent the mechanisms behind the decision-making processes that were found in both study arms. 

The study provided some indication for different types of GPs regarding their CDM, but the sample was not sufficiently large to be able to distinguish divergent types of CDM. However, it will be an interesting next step to assess whether different profiles of GPs can be distinguished, as has also been described by Roberts et al 2014, who described the three role archetypes GPs can fulfill while identifying mental health problems in children and youths: ‘fixers’, ‘future planners’ and ‘collaborators’. We elaborated on this point in the Discussion.

* Sampling technique: All GP´s were recruited from a regional academic research Network – this might have lead to selection effects (or all all GPs in the region members of this network)? 

Page 21. We agree the recruitment from the regional academic research network might have led to inclusion of GPs who were particularly interested in youth care. This may have influenced our results. Therefore, we have added this point to the Discussion.

* I did not fully understand the criteria for GPs who were approached for the quantitatve study: all remaining GPs not expressing their interest in research? Plus GPs not responding to the qualittive study?

It seems that GPs could indicate their interest in both studies, but there was no overlap between both study groups – was this by purpose or not?

Pages 8 & 11. All GPs who were registered in the research network were invited to participate in both the quantitative and qualitative substudy. Unpurposely, all GPs who participated in the qualitative substudy did not fill-out the questionnaire of the quantitative substudy. Both substudies were conducted at roughly the same time. We have processed our response in the text.

* Furthermore, if the qualitative study part included one participant max. per district, this would mean that the sample initially has been restricted to n=14 – or did the number and distribution happened by chance? If further inclusion/exclusion criteria like e.g. restricting the number of participants for the qualitative study based on the GP´s district, this should be made clear and should be justified.

Page 8. Since there are 14 districts in Holland Rijnland, the sample initially has been restricted to n=14. If two or more GPs from a particular district indicated their wish to participate in the qualitative substudy, the GP who responded first was included. We clarified our response in the text describing our sampling criteria.

* The sample obtained showed a long history of working as GPs; it might be of interest to see the range of years they have been working in the field. One would expect that GP´s with an average history of 18 years of practive usually have more routine in their CDM, and this might in part explain their hight self-rated competence.

Pages 11, 20 and Table 1. We have reported the range of years GPs have been working in the field and the range of years they have been working in their current general practice. We did not found that GPs who worked more years in the field had a higher self-rated competence. However, we found that more experienced GPs tend to give more priority to somatic instead of psychosocial problems. 

* Inclusion criteria (“GPs reporting seeing psychosocial problems among children and youth a minimum of three times per two weeks”) and finding in table 1 (max monthly psychosocial problems in children (35%) in the qualitative study and 50% in the quantitative study do not match. 

Page 9. We agree this inclusion criteria and these findings cause some confusion. In both substudies, we aimed to include GPs who see psychosocial problems among children and youths a minimum of three times per two weeks. However, during the recruitment phase, there were quite some GPs who reported seeing psychosocial problems about once to three times a month. Therefore, we have adjusted the sampling criteria. 

*Participants . Are the inclusion criteria defined as ALL criteria need to be fulfilled or at least one of the criteria has to be fulfilled?

Pages 8 & 9. The authors confirm that all sampling criteria needed to be fulfilled. As mentioned above, the criterion of seeing psychosocial problems among children and youths a minimum of three times per two weeks was found to be too strict, although it was our aim to include GPs who see these problems in a sufficient frequency.

---

## [Editor Report · Decision Letter 2]

15 Nov 2022

General practitioners’ everyday clinical decision-making on psychosocial problems of children and youth in the Netherlands

PONE-D-22-10593R2

Dear Dr. van Venrooij

We’re pleased to inform you that your manuscript has been judged scientifically suitable for publication and will be formally accepted for publication once it meets all outstanding technical requirements.

Kind regards,

Margaret Williams, Ph.D

Academic Editor

PLOS ONE
---

## [Editor Report · Acceptance letter]

16 Dec 2022

PONE-D-22-10593R2 

General practitioners’ everyday clinical decision-making on psychosocial problems of children and youth in the Netherlands 

Dear Dr. van Venrooij:

I'm pleased to inform you that your manuscript has been deemed suitable for publication in PLOS ONE. Congratulations! Your manuscript is now with our production department. 

Kind regards, 

on behalf of

Professor Margaret Williams 

Academic Editor

PLOS ONE